# Causal Bandits without prior knowledge using separating sets

**Arnoud A.W.M. de Kroon**                                      A.A.W.M.DEKROON@UVA.NL
*Korteweg-de Vries Institute*
*University of Amsterdam*

**Danielle Belgrave**                                      DANIBELLE25@GOOGLEMAIL.COM
*Work carried out while at Microsoft Research Cambridge*

**Joris M. Mooij**[*]                                      J.M.MOOIJ@UVA.NL
*Korteweg-de Vries Institute*
*University of Amsterdam*

**Editors:** Bernhard Schölkopf, Caroline Uhler and Kun Zhang

## Abstract

The Causal Bandit is a variant of the classic Bandit problem where an agent must identify the best action in a sequential decision-making process, where the reward distribution of the actions displays a non-trivial dependence structure that is governed by a causal model. Methods proposed for this problem thus far in the literature rely on exact prior knowledge of the full causal graph. We formulate new causal bandit algorithms that no longer necessarily rely on prior causal knowledge. Instead, they utilize an estimator based on separating sets, which we can find using simple conditional independence tests or causal discovery methods. We show that, given a true separating set, for discrete i.i.d. data, this estimator is unbiased, and has variance which is upper bounded by that of the sample mean. We develop algorithms based on Thompson Sampling and UCB for discrete and Gaussian models respectively and show increased performance on simulation data as well as on a bandit drawing from real-world protein signaling data.

## 1. Introduction

In recent years, there have been several works on the Causal Bandit problem (Lattimore et al., 2016; Sen et al., 2017; Yabe et al., 2018; Lee and Bareinboim, 2018). This is a variant of the classical multi-armed bandits problem, where we assume there is an underlying structural causal model (Pearl, 2009) between observed variables.

In the classical bandit problem, we iteratively select one arm from a set of arms, and then observe a reward variable conditional on the chosen arm, where the rewards for the arms are assumed to be independent. In the Causal Bandit problem, after we select an arm, we not only observe a reward variable but also some additional variables. The reward and these additional variables are generated according to a causal model, where each arm corresponds to a certain intervention on the causal model. In this case, the rewards of the arms are no longer independent, and the previous works on this topic have shown that one can use this additional structure to improve performance. (Lattimore et al., 2016; Sen et al., 2017; Yabe et al., 2018; Lee and Bareinboim, 2018)

Recent approaches to the Causal Bandit problem have shown greatly improved bounds for regret compared to naïve approaches. This is done either by using information leakage, where data from

---

[*] JMM was supported by the European Research Council (ERC) under the European Union's Horizon 2020 research and innovation programme (grant agreement 639466).

one arm is used to improve the estimates of other arms (Lattimore et al., 2016; Sen et al., 2017; Yabe et al., 2018; Lu et al., 2020), or by pruning arms for which it can be concluded from the structure of the causal graph that they can not be the optimal arm (Lee and Bareinboim, 2018). However, they all rely on perfect prior knowledge of the causal structure. The works using information leakage also assume that interventional distributions on variables beside the target variable are known beforehand.

Because in practice this information may often not be available, in this work, we develop a framework for Causal Bandits where we try to learn everything from scratch. We build an estimator using the *separating set* concept known from the causality literature (Spirtes et al., 2000; Magliacane et al., 2018; Rojas-Carulla et al., 2018). This is a set $\mathbf{S}$ that has the property that it renders a target variable $Y$ independent of a context variable $I$ when conditioned upon (which we denote $I \perp\!\!\!\perp Y \mid \mathbf{S}$), where the context variable encodes which interventions are performed. Contrary to previously used estimators, this property can be estimated from Causal Bandit data directly by conditional independence tests, and does not require a full causal discovery method. However, causal discovery methods may still provide a benefit compared to just conditional independence tests if they can efficiently combine different conditional independence tests results to correct for errors.

Our contributions in this work are as follows:

- We propose a new algorithm for discrete Causal Bandits based on Thompson Sampling as well as an algorithm for linear Gaussian Causal Bandits with discrete soft interventions based on UCB-normal, where no prior knowledge is assumed of the Causal Bandit.

- These algorithms use an estimator based on the Separating Set property, which allows the algorithm to work in presence of cycles or confounding, as the Separating Set is still a valid notion in these graphs. We can directly test this property from data, and this property can also be inferred from the output of causal discovery algorithms. This allows us to drop the assumption of prior causal knowledge.

- We show that for discrete variables even without prior knowledge of interventional distributions, if we assume that a true separating set is known a-priori, the used estimator has variance upper bounded by the variance of a naïve sample mean estimator.

- For discrete variables we show greatly improved cumulative regret performance compared to classical bandit algorithms in simulation studies on graphs where a separating set indeed exists but is not known a-priori. We apply the linear Gaussian model variant of our algorithm to a Causal Bandit where data is generated by pulling from the dataset presented in Sachs et al. (2005), which is experimental data concerning protein-signaling networks. Here we again show improved performance compared to the traditional UCB-normal algorithm.

While the experiments in this work are only for small graphs, these contributions show that significant performance improvements are possible if the underlying Causal Bandit has separating sets, even in the case where no prior causal knowledge is available.

## 2. Preliminaries

In this section we introduce the required preliminaries regarding causality and Causal Bandits.

## 2.1. Causal modeling and graph definitions

We will very briefly introduce the elements of the theory of graphical causal modeling that are used in this work. An in-depth introduction can for example be found in Pearl (2009).

We will denote tuples of variables with a bold capital letter, e.g. $\mathbf{X} = (X_i)_{i=1}^{n}$, and will use lower case letter $x$ for a value assigned to $X$. The domain of $\mathbf{X}$ is denoted by $D(\mathbf{X})$. We assume that we observe variables generated through an acyclic Structural Causal Model $\mathcal{M} = \langle \mathbf{V}, \mathbf{E}, \mathbf{F}, \mathbb{P}[\mathbf{E}] \rangle$, which contains a tuple of endogenous variables $\mathbf{V}$ and a tuple of independent exogenous variables $\mathbf{E}$, where the latter has associated probability distribution $\mathbb{P}[\mathbf{E}]$. The values of $\mathbf{V}$ are defined by the tuple of functions $\mathbf{F}$, where for each $V_i \in \mathbf{V}$ there is a $f_{V_i} \in \mathbf{F}$ such that $V_i = f_{V_i}(pa(V_i), \mathbf{E}_i)$. Here $pa(V_i) \subseteq \mathbf{V} \setminus \{V_i\}$ are the direct causes ("parents") of $V_i$ and $\mathbf{E}_i \subset \mathbf{E}$ is a subset of the exogenous variables. We explicitly allow for confounders (since the $\mathbf{E}_i$ can overlap), but exclude cycles for simplicity of exposition though it would be straightforward to include them (see e.g. Mooij et al., 2020).

Each SCM has an associated graph $G = \langle \mathbf{V}, \mathcal{E} \rangle$, which is acyclic if and only if the SCM is acyclic, where $\mathbf{V}$ is a set of nodes corresponding to the endogenous variables and $\mathcal{E}$ is a set of edges. If $V_i$ directly influences $V_j$ according to $f_{V_j}$, then there is a directed edge $V_i \to V_j \in \mathcal{E}$. There is a bidirected edge $V_i \leftrightarrow V_j \in \mathcal{E}$ if they share independent noise variables, i.e., if $\mathbf{E}_i \cap \mathbf{E}_j \neq \emptyset$. We adopt the default family relationships: $pa, ch, an,$ and $de$ for parents, children and ancestors and descendants respectively, where for $an$ and $de$ we include the variable itself.

We may now reason about performing interventions on the variables $V_i$. In the SCM causal modeling framework, interventions are defined by altering the functional dependencies of the SCM. For example, we may force the value of a variable to a specific value $\xi$. This is called a *perfect intervention*, and the joint probability is then notated as $\mathbb{P}[\mathbf{V} \,|\, \mathrm{do}(V_i = \xi)]$. One may also define other types of interventions, for example *soft* interventions which alter the functional dependency $f_{V_i}$ but may keep a functional relationship on its parents, as opposed to just setting the variable to a value.

In this work we make use of *context variables* as an elegant way to model interventions, as found in for example Spirtes et al. (2000); Pearl (2009); Mooij et al. (2020); Dawid (2021). We introduce $\mathbf{I}$ to be the set of context variables. We will consider graphs $\mathcal{G} = (\mathbf{V} \cup \mathbf{I}, \mathcal{E})$ with additional vertices $\mathbf{I}$ corresponding to different interventions. If $I_i \in \mathbf{I}$ encodes an intervention on nodes $\mathbf{T}_i \subseteq \mathbf{V}$, we set $I_i$ to $\varnothing$ if we do not perform this intervention, and to a different value $\xi$ for each possible version of intervention $I_i$. Furthermore, we add an edge $I_i \to V_i$ to $\mathcal{E}$ for each $V_i \in \mathbf{T}_i$. For example, we can model a perfect intervention $\mathrm{do}(V_i = \zeta)$ by intervention variable $I_i$ if we modify $f_{V_i}$ to:

$$f_{V_i}^*(pa^*(V_i), \mathbf{E}_i) = \begin{cases} \zeta & \text{if } I_i = \zeta \\ f_{V_i}(pa(V_i), \mathbf{E}_i) & \text{if } I_i = \varnothing \end{cases}$$

with $pa^*(V_i) = Pa(V_i) \cup \{I_i\}$. Then, if we perform some combination of interventions, this corresponds to choosing a vector of values $\boldsymbol{\zeta}$, of the same size as the number of intervention variables, and where some values may be $\varnothing$, resulting in $\mathbb{P}[\mathbf{V} \,|\, \mathrm{do}(\mathbf{I} = \boldsymbol{\zeta})]$. Note that with this formalism, $\mathbb{P}[\mathbf{V} \,|\, \mathrm{do}(\mathbf{I} = \boldsymbol{\zeta})] = \mathbf{P}[\mathbf{V} \,|\, \mathbf{I} = \boldsymbol{\zeta}]$, because the intervention variables are exogenous.

We define a *path* between nodes $V_0$ and $V_n$ as a tuple $\langle V_0, e_1, V_1, e_2, \ldots, e_n, V_n \rangle$, with $V_i \in \mathbf{V}$, $e_i \in \mathcal{E}$, where each node occurs at most once and $e_i$ is an edge of type $\leftarrow, \to,$ or $\leftrightarrow$ with endpoints $V_{i-1}$ and $V_i$. $V_k$ is called a *collider* on a path if there is a subpath $\langle V_{k-1}, e_k, V_k, e_{k+1}, V_{k+1} \rangle$ where

the edges $e_k$ and $e_{k+1}$ meet head to head on node $V_k$ (i.e. both edges have an arrowhead into $V_k$). Otherwise this node is called a *non-collider*. The endpoints are also refered to as non-colliders.

Using the definition of paths and colliders, one defines d-separation:

**Definition 1** (d-*separation*) *We say a path* $\langle V_0, e_1, \ldots, e_n, V_n \rangle$ *in graph* $\mathcal{G} = (\mathbf{V}, \mathcal{E})$ *is blocked by* $\mathbf{C} \subseteq \mathbf{V}$ *if:*
*(i): Its first or last node is in* $\mathbf{C}$*, or*
*(ii): It contains a collider on a node not in* $an(\mathbf{C})$*, or*
*(iii): It contains a non-collider in* $\mathbf{C}$
*If for sets* $\mathbf{A}, \mathbf{B} \subseteq \mathbf{V}$ *all paths from nodes in* $\mathbf{A}$ *to nodes in* $\mathbf{B}$ *are blocked by* $\mathbf{C} \subseteq \mathbf{V}$*, we say that* $\mathbf{A}$ *is d-separated from* $\mathbf{B}$ *by* $\mathbf{C}$*, and write* $\mathbf{A} \perp_{\mathcal{G}} \mathbf{B} \,|\, \mathbf{C}$*.*

Consider an acyclic SCM $\mathcal{M}$ with graph $\mathcal{G}$. Let $\mathbb{P}_{\mathcal{M}}$ be the probability distribution induced by this model. Then the Directed Global Markov Property holds for subsets $\mathbf{A}, \mathbf{B}, \mathbf{C} \subseteq \mathbf{V}$:

$$\mathbf{A} \perp_{\mathcal{G}} \mathbf{B} \,|\, \mathbf{C} \implies \mathbf{A} \perp\!\!\!\perp_{\mathbb{P}_{\mathcal{M}}} \mathbf{B} \,|\, \mathbf{C}.$$

These conditional independencies are the core information provided by causal reasoning that we exploit in this work. While our algorithm itself does not explicitly assume the converse (called *faithfulness*), this is assumed by many causal discovery algorithms thus we henceforth assume faithfulness as well.

### 2.2. Causal Bandit problem

The multi-armed bandit problem is one of the classic problems studied in sequential decision making literature (Lai and Robbins, 1985). In this setting, an agent decides on which arm to pull and receives a reward corresponding to that arm. Classically, the rewards of the arms are considered independent which gives rise to strategies like $\varepsilon$-greedy, UCB (Auer et al., 2002; Cappé et al., 2013) and Thompson Sampling (Thompson, 1933).

Lattimore, Lattimore, and Reid (2016) introduced the Causal Bandit problem as follows. Consider an agent in a sequential decision making process consisting of $T$ trials. In each trial, the agent chooses an assignment of values $\boldsymbol{\zeta}$ to intervention variables $\mathbf{I}$ (also referred to as choosing an arm). It then observes variables from $\mathbb{P}[\mathbf{V} \,|\, \mathbf{I} = \boldsymbol{\zeta}]$, according to an SCM $\mathcal{M} = \langle \mathbf{V}, \mathbf{E}, \mathbf{I}, \mathbf{F}, \mathbb{P}[\mathbf{E}] \rangle$ with corresponding graph $\mathcal{G} = (\mathbf{V} \cup \mathbf{I}, \mathcal{E})$. One of the endogenous variables $Y \in \mathbf{V}$ is the target variable. Thus, when choosing an arm for trial $N + 1$, the agent has observed data $\mathcal{D}^N = \{(\boldsymbol{\zeta}^n, \mathbf{v}^n)\}_{n=1}^N$, which are pairs of intervention node values $\boldsymbol{\zeta}$ and realizations of $\mathbf{V}$. Let $Y^n$ denote the target variable observed in trial $n$. The goal is then to minimize the cumulative regret $\mathcal{R} = \sum_{n=1}^T [Y^n - \max_{\boldsymbol{\zeta}} \mathbb{E}[Y \,|\, \mathbf{I} = \boldsymbol{\zeta}]]$.

For convenience, we will introduce notation to count the number of samples in our data for which a certain predicate $p$ holds. Let $\mathcal{N}_{\mathcal{D}^N}(p) = |\{(\boldsymbol{\zeta}^n, \boldsymbol{v}^n) \in \mathcal{D}^N \,|\, (\boldsymbol{\zeta}^n, \boldsymbol{v}^n) \vDash p\}|$. For example, $\mathcal{N}_{\mathcal{D}^N}(Y = 1, \mathbf{I} = \boldsymbol{\zeta})$ is the number of samples in dataset $\mathcal{D}^N$ for which we performed intervention $\boldsymbol{\zeta}$ and observed the value 1 for reward variable $Y$.

### 2.3. Related Work

The algorithms proposed for the Causal Bandit problem can be broadly subdivided into two categories: (i) those relying on information leakage and (ii) those that prune the action space based on the structure

of the causal graph. The initial paper by Lattimore et al. (2016) gave improved bounds for simple regret for the Causal Bandit problem. This was done by utilizing *information leakage*: the reward obtained under one intervention may provide information about other interventions. The authors construct an importance sampling estimator based on this principle that assumes full prior knowledge of the probability distribution of all variables besides the target variable. Using this, the authors derive an improved simple regret bound. Sen et al. (2017) focused on applying more advanced techniques from the Bandit literature. For example, they analyze gap dependent bounds and apply dynamic clipping, where they divide the $T$ trials into phases and apply a different clipping constant for each phase. These advances lead to sometimes exponentially better regret than the algorithm by Lattimore et al. (2016).

Yabe et al. (2018) extend Lattimore et al.'s work in a different direction. They consider only binary variables and perfect interventions on subsets of nodes. They use the full knowledge of the graph to estimate the probabilities $p(V \mid pa(V), \mathbf{I} = \boldsymbol{\zeta})$ for each node $V \in \mathbf{V}$. Interestingly, they only require prior knowledge of the graph and estimate all required probability distributions from data acquired from the actual bandit. By modeling the relation of each individual node with regards to its parents, if we have two data sources which only differ by the intervention value of a single node $V_i$ (i.e. all other nodes are intervened on the same way in both data sources), we can still update all estimated probability distributions to be updated for $V_j \neq V_i$, thereby allowing information leakage between different interventions.

More recently, Lee and Bareinboim (2018) introduced a new method for the Causal Bandit problem. They consider perfect interventions on subsets of nodes of the causal graph. Because they only consider perfect interventions, it is sometimes impossible for some interventions to always perform at least as well as some other interventions more upstream, and thus these upstream interventions may be pruned.

One thing that all previously discussed approaches have in common is that they assume the causal relationships to be known beforehand, an assumption that is often not met in practice. Very recently, Lu, Meisami, and Tewari (2021) base a causal bandit algorithm on this where the graph is not known beforehand, though this approach is limited to perfect interventions on all subsets.

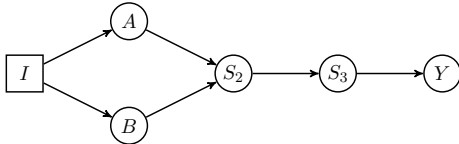

Figure 1: Example causal graph $\mathcal{G}$. $I$ is an intervention variable that encodes interventions on $A$ and $B$, $Y$ is the reward variable. Possible choices for separating set $\mathbf{S}$ for sharing data between interventions on $A$ and $B$ are $\{A, B\}, \{S_2\}, \{S_3\}$, and any superset of these sets, since then $I \perp_{\mathcal{G}} Y \mid \mathbf{S}$.

## 3. Separating sets lead to improved estimators

In this section, we generalize all previous works on Causal Bandits using information leakage that rely on a-priori knowledge of the full causal structure, and propose a more general estimator based on separating sets. This estimator does not rely on a-priori known distributions, and has favorable properties compared to direct sample mean estimation. In this section we assume that a

true separating set is given. In our algorithm and experiment sections we search for separating sets using causal discovery or conditional independence tests.

### 3.1. The information sharing estimator

A core strategy we have seen in Causal Bandits in previous works is to exploit very specific knowledge about the causal structure in order to construct estimators that share information between arms. In order to make Causal Bandits more easily suitable for causal discovery, we introduce a novel information sharing estimator that relies on much less specific knowledge about the causal graph, but may still exploit information leakage to share data between interventions.

We say that a set of variables $\mathbf{S}$ is a *separating set* for intervention variables $\mathbf{I}$ and target variable $Y$ if $\mathbf{I} \perp_{\mathcal{G}} Y \,|\, \mathbf{S}$. By the Markov property and faithfulness, this is equivalent to the conditional independence $\mathbf{I} \perp\!\!\!\perp_{\mathbb{P}_{\mathcal{M}}} Y \,|\, \mathbf{S}$.

If $\mathbf{S}$ is a separating set, we have for all possible interventions $do(\mathbf{I} = \boldsymbol{\zeta})$ the following identity by the law of total expectation, where the second equality uses the independence:

$$\mathbb{E}[Y \,|\, \mathbf{I} = \boldsymbol{\zeta}] = \mathbb{E}[\mathbb{E}[Y \,|\, \mathbf{S}, \mathbf{I} = \boldsymbol{\zeta}] \,|\, \mathbf{I} = \boldsymbol{\zeta}] \tag{1}$$
$$= \mathbb{E}[\mathbb{E}[Y \,|\, \mathbf{S}] \,|\, \mathbf{I} = \boldsymbol{\zeta}].$$

This relationship underlies previous works on information leakage, where $\mathbb{P}[\mathbf{S}|\mathbf{I} = \boldsymbol{\zeta}]$ is assumed known beforehand for some fixed choice of $\mathbf{S}$ which is always a valid separating set under the assumptions made in the relevant work. For example, in case unobserved confounding is ruled out as is done in Lattimore et al. (2016), the parents of $Y$ are always a valid separating set. Under these strict assumptions, the only unknown we have to estimate is $\mathbb{E}[Y \,|\, \mathbf{S}]$, whereby we see that in the cumulative regret upper bound the $\sqrt{|D(\mathbf{I})|}$ term (where $D(\mathbf{I})$ corresponds to the number of arms) is reduced to $\sqrt{|D(\mathbf{S})|}$ (Lu et al., 2020), i.e. the dimensionality of the arms is essentially reduced from $D(\mathbf{I})$ to $D(\mathbf{S})$. Formulating the estimator in terms of separating sets as we do in this work is much less restrictive than specifically focussing on e.g. the parents of $Y$, as in case of cycles or confounding the parents of $Y$ are not a separating set in general, in which case there may still be other groups of variables that satisfy the separating set property.

Instead we focus on the case where there are no priorly known distributions. We introduce separate estimators $\hat{\mu}(y \,|\, \mathbf{s}; \mathcal{D}^N)$ for $\mathbb{E}[Y \,|\, \mathbf{S} = \mathbf{s}]$ and $\hat{p}(\mathbf{s} \,|\, \boldsymbol{\zeta}, \mathcal{D}^N)$ for $\mathbb{P}[\mathbf{S} = \mathbf{s} \,|\, \mathbf{I} = \boldsymbol{\zeta}]$. Inspired by the above identity, we then propose the following *information sharing estimator* for $\mathbb{E}[Y \,|\, \mathbf{I} = \boldsymbol{\zeta}]$:

$$\hat{\mu}_{IS}(y \,|\, \boldsymbol{\zeta}; \mathcal{D}^N, \mathbf{S}) := \sum_{\mathbf{s} \in D(\mathbf{S})} \hat{\mu}(y \,|\, \mathbf{s}; \mathcal{D}^N) \hat{p}(\mathbf{s} \,|\, \boldsymbol{\zeta}, \mathcal{D}^N). \tag{2}$$

If $\mathbf{S}$ is discrete and $Y$ is binary, an obvious choice is to use the empirical conditional distribution for $\hat{p}(\mathbf{s} \,|\, \boldsymbol{\zeta}, \mathcal{D}^N)$ and the sample mean for $\hat{\mu}(y \,|\, \mathbf{s}; \mathcal{D}^N)$, i.e. the maximum likelihood estimators. If we are dealing with linear Gaussian models where the interventions are still discrete, we again use the empirical conditional distribution for $\hat{p}(\mathbf{s} \,|\, \boldsymbol{\zeta}, \mathcal{D}^N)$ and a linear regression model for $\hat{\mu}(y \,|\, \mathbf{s}; \mathcal{D}^N)$. For now we focus on discrete variables with binary target variable $Y$, and thus we define:

$$\hat{p}(\mathbf{s}|\boldsymbol{\zeta}, \mathcal{D}^N) := \frac{\mathcal{N}_{\mathcal{D}^N}(\mathbf{S} = \mathbf{s}, \mathbf{I} = \boldsymbol{\zeta})}{\mathcal{N}_{\mathcal{D}^N}(\mathbf{I} = \boldsymbol{\zeta})}, \tag{3}$$

$$\hat{\mu}(y \,|\, \mathbf{s}; \mathcal{D}^N) := \frac{\mathcal{N}_{\mathcal{D}^N}(Y = 1, \mathbf{S} = \mathbf{s})}{\mathcal{N}_{\mathcal{D}^N}(\mathbf{S} = \mathbf{s})}. \tag{4}$$

This estimator has some nice properties. To estimate $\hat{p}(\mathbf{s}|\boldsymbol{\zeta}, \mathcal{D}^N)$, we use only data gathered under a specific intervention. However, to estimate $\hat{\mu}(\mathbf{s}; \mathcal{D}^N)$ we may *share* data across interventions, which leads to reduced variance. Suppose data pooling is not possible, for example because we only have data gathered under a specific intervention available, then $\mathcal{N}_{\mathcal{D}^N}(\mathbf{S} = \mathbf{s}) = \mathcal{N}_{\mathcal{D}^N}(\mathbf{S} = \mathbf{s}, \mathbf{I} = \boldsymbol{\zeta})$ and $\mathcal{N}_{\mathcal{D}^N}(Y = 1, \mathbf{S} = \mathbf{s}) = \mathcal{N}_{\mathcal{D}^N}(Y = 1, \mathbf{S} = \mathbf{s}, \mathbf{I} = \boldsymbol{\zeta})$. The information sharing estimator then reduces nicely to just the naïve direct sample mean $\hat{\mu}_{SM}(\boldsymbol{\zeta}; \mathcal{D}^N)$ (note this result also holds in the Gaussian additive noise model case):

$$\hat{\mu}_{IS}(y \,|\, \boldsymbol{\zeta}; \mathcal{D}^N, \mathbf{S}) = \sum_{\mathbf{s} \in D(\mathbf{S})} \frac{\mathcal{N}_{\mathcal{D}^N}(Y = 1, \mathbf{S} = \mathbf{s}, \mathbf{I} = \boldsymbol{\zeta})}{\mathcal{N}_{\mathcal{D}^N}(\mathbf{I} = \boldsymbol{\zeta})},$$
$$= \frac{\mathcal{N}_{\mathcal{D}^N}(Y = 1, \mathbf{I} = \boldsymbol{\zeta})}{\mathcal{N}_{\mathcal{D}^N}(\mathbf{I} = \boldsymbol{\zeta})} = \hat{\mu}_{SM}(\boldsymbol{\zeta}; \mathcal{D}^N).$$

However, if pooling *is* possible, we expect our estimator to outperform naïve estimation, because it uses more data in an efficient manner. Indeed, in the appendix we show that the following theorem holds:

**Theorem 2** *If we calculate $\hat{\mu}_{IS}(y \,|\, \boldsymbol{\zeta}; \mathcal{D}^N, \mathbf{S})$ from a dataset $\mathcal{D}^N$ where we have a fixed number of i.i.d. data points from each possible intervention generated by a discrete Causal Bandit, and $\mathbf{I} \perp\!\!\!\perp_{\mathbb{P}_{\mathcal{M}}} Y \,|\, \mathbf{S}$, and there is at least one sample from intervention $\mathbf{I} = \boldsymbol{\zeta}$, then the information sharing estimator (2) is unbiased. Furthermore, its variance is upper bounded by that of the sample mean:*

$$\mathbb{V}[\hat{\mu}_{IS}(y \,|\, \boldsymbol{\zeta}; \mathcal{D}^N, \mathbf{S})] \le \mathbb{V}[\hat{\mu}_{SM}(y \,|\, \boldsymbol{\zeta}; \mathcal{D}^N)] \tag{5}$$

**Proof** See appendix. ∎

In the proof we can see that the variance of the information sharing estimator can be seen as a decomposition of the sample mean variance into two terms. The first term arises from misestimation of $\mathbb{P}[\mathbf{S} \,|\, \mathbf{I} = \boldsymbol{\zeta}]$, which can only be reduced by adding more data where $\mathbf{I} = \boldsymbol{\zeta}$. The second term arises from misestimation of $\mathbb{E}[Y \,|\, \mathbf{S}]$, which *can* be reduced by data where $\mathbf{I} \ne \boldsymbol{\zeta}$, as long as $\mathbb{P}[\mathbf{S} \,|\, \mathbf{I} = \boldsymbol{\zeta}]$ and $\mathbb{P}[\mathbf{S} \,|\, \mathbf{I} \ne \boldsymbol{\zeta}]$ do not have disjoint support. In the appendix we provide a more thorough analysis of the variance.

### 3.2. Selecting the best separating set

It is important to note that it is not clear from just the structure of the causal graph which separating set leads to the best estimator in case there are multiple such sets available. Therefore, even in the case of a known causal graph, fixing the chosen separating set is not ideal. Consider the graph from figure (1). In previous works with improved regret bounds using information leakage, either $\{A, B\}$ (Sen et al., 2017) or $S_3$ (Lattimore et al., 2016; Lu et al., 2020) would be the fixed choice for separating set. However, for each of the three choices of a minimal separating set there exist SCM's for which this choice leads to an estimator that has strictly smaller variance than when we choose one of the other two separating sets. For example, consider the case where the associated SCM is defined by:

$$\begin{cases} A = I, & S_3 = S_2 \text{ XOR } U_2, \\ B = 1 - I, & Y = S_3, \\ S_2 = I_A \text{ XOR } I_B \text{ XOR } U_1, \end{cases}$$

and data gathered is from both intervention $do(I = 0)$ and intervention $do(I = 1)$. In this case, if we select $\{S_3\}$ as separating set, then since $S_3$ is a copy of $Y$, our estimator for $\mathbb{E}[Y|S_3]$ will be perfect after just one sample of each possible value of $S_3$ and thus its error cannot be reduced by information sharing. In this case, the variance of the information sharing estimator is exactly the same as that of the sample mean. If we select $S_1 = \{A, B\}$ the support of the two interventional distributions on $S_1$ will be disjoint, and thus information sharing does not lead to an improved estimate and again reduces to the sample mean. If we select $\{S_2\}$ however, information sharing is possible since there is overlap on $\mathbb{P}[S_2 \mid I = 1]$ and $\mathbb{P}[S_2 \mid I = 0]$ and more samples actually reduce our error in the estimation of $\mathbb{E}[Y|S_2]$ since the relationship between $Y$ and $S_2$ is not deterministic, and our estimator will have reduced variance. Similarly, we may construct cases where both of the other sets are strictly optimal.

Thus, given multiple candidate separating sets it is not immediately obvious which one we should pick. The strategy we use in this paper is to estimate the variance for each candidate separating set and pick the one with the lowest estimated variance.

### 3.3. Partial separating sets

So far, for ease of exposition, we have implicitly assumed that we always find separating sets that are separating for *all* intervention variables. In practice this may not be the case. Let us consider the case where we have a subset $\mathbf{I}' \subset \mathbf{I}$ and a *partial* separating set $\mathbf{S}$ such that $\mathbf{I}' \perp_{\mathcal{G}} Y \mid \mathbf{S}$, i.e. it may not be separating for all intervention variables. Let $\mathbf{I}_{NS} = \mathbf{I} \setminus \mathbf{I}'$ and $\boldsymbol{\zeta}_{\mathbf{I}_{NS}}$ be $\boldsymbol{\zeta}$ restricted to $\mathbf{I}_{NS}$. We may then use a similar decomposition as before:

$$\mathbb{E}[Y \mid \mathbf{I} = \boldsymbol{\zeta}] = \mathbb{E}[\mathbb{E}[Y \mid \mathbf{S}, \mathbf{I} = \boldsymbol{\zeta}] \mid \mathbf{I} = \boldsymbol{\zeta}]$$
$$= \mathbb{E}[\mathbb{E}[Y \mid \mathbf{S}, \mathbf{I}_{NS} = \boldsymbol{\zeta}_{\mathbf{I}_{NS}}] \mid \mathbf{I} = \boldsymbol{\zeta}],$$

where we use that $\mathbf{I}' \perp_{\mathcal{G}} Y \mid \mathbf{S} \implies \mathbf{I}' \perp_{\mathcal{G}} Y \mid \mathbf{S} \cup \mathbf{I}_{NS}$, since the context variables have no parents and thus they can not be a collider and conditioning on them can not open a path. Motivated by this, we define the slightly more refined information sharing estimator:

$$\hat{\mu}_{IS}(y \mid \boldsymbol{\zeta}; \mathcal{D}^N, \mathbf{S}, \mathbf{I}_{NS}) := \sum_{\mathbf{s} \in D(\mathbf{S})} \hat{\mu}(y \mid \mathbf{s}; \mathcal{D}^N, \mathbf{I}_{NS}) \hat{p}(\mathbf{s} \mid \boldsymbol{\zeta}, \mathcal{D}^N), \tag{6}$$

where to estimate $\hat{\mu}(y \mid \mathbf{s}; \mathcal{D}^N, \mathbf{I}_{NS})$ we only pool data where $\mathbf{I}_{NS} = \boldsymbol{\zeta}_{\mathbf{I}_{NS}}$ (i.e. $\mathbf{I}'$ may take any value). For discrete models we then define:

$$\hat{\mu}(y \mid \mathbf{s}; \mathcal{D}^N, \mathbf{I}_{NS}) := \frac{\mathcal{N}_{\mathcal{D}^N}(Y = 1, \mathbf{S} = \mathbf{s}, \mathbf{I}_{NS} = \boldsymbol{\zeta}_{\mathbf{I}_{NS}})}{\mathcal{N}_{\mathcal{D}^N}(\mathbf{S} = \mathbf{s}, \mathbf{I}_{NS} = \boldsymbol{\zeta}_{\mathbf{I}_{NS}})},$$

where for intervention $\boldsymbol{\zeta}$ we pool data over all possible values for $\mathbf{I}'$. For linear Gaussian models, we fit $\hat{\mu}(\mathbf{s}; \mathcal{D}^N, \mathbf{I}_{NS})$ using data that is pooled in the exact same manner.

### 4. Separating Set Causal Bandit Algorithms

Given the a separating set, for discrete variables with binary target variable, it is natural to define a Separating Set Causal Bandit algorithm based on Thompson sampling. We model the parameters $\mathbb{P}[\mathbf{S} = \mathbf{s}|\mathbf{I} = \boldsymbol{\zeta}]$ using a Dirichlet prior and the parameters $\mathbb{P}[Y = 1|\mathbf{S} = \mathbf{s}]$ using a Beta prior. We can then apply Thompson sampling, by sampling the parameters from their posterior distributions,

---

**Algorithm 1** Causal TS

---

1: **Input:** Data: $\mathcal{D}^N = \{(\boldsymbol{\zeta}^n, \boldsymbol{v}^n)\}_{n=1}^N$, set of possible interventions $D(\mathbf{I})$, target variable $Y$, number of MC samples $n_{MC}$, Separating set algorithm: $disc\_sep\_set$

2: **Output:** Next action to pick at iteration $N + 1$

3: Initialize array $index[\boldsymbol{\zeta}]$ **for** $\boldsymbol{\zeta} \in D(\mathbf{I})$

4: $\mathbf{S\_set} \leftarrow disc\_sep\_set(\mathcal{D}^N, Y, \boldsymbol{\zeta})$

5: **for all** $\boldsymbol{\zeta} \in D(\mathbf{I})$ **do**

6:     Sample $\tilde{\mu} \sim Beta(\mathcal{N}_{\mathcal{D}^N}(Y = 1, \mathbf{I} = \boldsymbol{\zeta}) + 1, \mathcal{N}_{\mathcal{D}^N}(Y = 0, \mathbf{I} = \boldsymbol{\zeta}) + 1)$

7:     $index[\boldsymbol{\zeta}] = \tilde{\mu}$

8:     $best\_var \leftarrow \mathbb{V}[\tilde{\mu}]$

9:     **for all** $(\mathbf{S}, \mathbf{I}') \in \mathbf{S\_set}$ **do**

10:         $\mathbf{I}_{NS} = \mathbf{I} \setminus \mathbf{I}'$

11:         Initialize array $samples[i]$ for $i \in \{1, \ldots, n_{MC}\}$

12:         **for** $i$ in $\{1, \ldots, n_{MC}\}$ **do**

13:             Sample $\tilde{\mathbf{p}} \sim Dirichlet(\{\mathcal{N}_{\mathcal{D}^N}(\mathbf{S} = \mathbf{s}, \mathbf{I} = \boldsymbol{\zeta}) + 1\}_{\mathbf{s} \in D(S)})$

14:             Sample $\tilde{\boldsymbol{\mu}} \sim (Beta(\mathcal{N}_{\mathcal{D}^N}(Y = 1, \mathbf{S} = \mathbf{s}, \mathbf{I}_{NS} = \boldsymbol{\zeta}_{NS}) + 1,$
                              $\mathcal{N}_{\mathcal{D}^N}(Y = 0, \mathbf{S} = \mathbf{s}, \mathbf{I}_{NS} = \boldsymbol{\zeta}_{NS}) + 1))_{\mathbf{s} \in D(\mathbf{S})}$

15:             $samples[i] \leftarrow \tilde{\mathbf{p}}^T \tilde{\boldsymbol{\mu}}$

16:         **end for**

17:         **if** $\hat{\mathbb{V}}[samples] < best\_var$ **then**

18:             $index[\boldsymbol{\zeta}] = samples[1]$

19:             $best\_var = \hat{\mathbb{V}}[samples]$

20:         **end if**

21:     **end for**

22: **end for**

23: **return** $\arg\max_{\boldsymbol{\zeta}} index[\boldsymbol{\zeta}]$

---

and calculating the resulting expected value of $Y$ as if these were the true parameters (using equation 1). Given a sample $(\boldsymbol{\zeta}, \mathbf{s}, Y)$, we can update each of the posteriors separately and naturally.

We assume that we have a causal discovery algorithm that has a function $disc\_sep\_set$, which given a dataset and target variable, returns a set of pairs that maps subsets $\mathbf{S}$ of the endogenous variables $\mathbf{V} \setminus \{Y\}$, to the set of interventions that are separated by this set. If this set is not empty, $\mathbf{S}$ is at least a partial separating set which may then be used to construct an information sharing estimated. If we have multiple choices of separating sets, we choose the set where the posterior model has the lowest variance with regard to the sample mean according to equation 1. The variance of the sample mean may easily be estimated using a Monte Carlo estimation since all parameters are known. We compare this variance to that of the naïve Thompson Sampling model, and revert to it if no improvement is found.

The full algorithm is provided in Algorithm 1. On line 4 we run causal discovery. In our experiments to save computation we only run this every time the number of iterations increases by 25%. For each possible assignment of $\zeta$, on line 6-8 we first initialize variables that correspond to the index and estimated variance of the best model found so far, and initialize these to those corresponding to a traditional Thompson Sampling model without information sharing. On lines 11-17, for each partial separating set we estimate the variance of the model resulting from choosing

this separating set, and pick as index the lowest variance model on line 18, where we thus fall back on the traditional Thompson Sampling baseline if no information sharing model is found that improves on its variance.

For Gaussian additive noise models, we adapt the traditional UCB-normal algorithm for unknown variance (Auer et al., 2002). The only changes we make to this algorithm is the use of the information sharing estimator (with linear regression models for $\hat{\mu}$) instead of the normal sample mean, and bootstrapping to obtain an estimate of the variance. Full details of this algorithm as well as the baseline are in the appendix in algorithm 2.

We may combine our novel algorithm with any causal discovery algorithm which outputs separating sets. The methods we used in our experiments are described in the following subsections.

### 4.1. Direct Independence Testing

Since we have full interventional data, we may simply directly test for all sets $\mathbf{S}$ whether they have the separating set property, i.e., whether $\mathbf{I} \perp\!\!\!\perp_{\mathbb{P}_{\mathcal{M}}} Y \,|\, \mathbf{S}$. Our baseline causal discovery method is then to directly test for separating sets from data in this way. For discrete variables, we make use of the $G^2$-test for conditional independence of discrete variables (Neapolitan, 2004) with $p$-value threshold $\alpha = 0.05$. For linear Gaussian models, we make use of an independence test from step 1 in section 3.1.2 in Peters et al. (2016) with a p-value treshold of $\alpha = 0.05$.

### 4.2. ASD-JCI123kt

A state-of-the-art causal discovery algorithm for small numbers of variables is ASD-JCI123kt (Mooij et al., 2020). It is a particular implementation of the Joint Causal Inference framework (Mooij et al., 2020), which pools data over contexts. This allows it to simultaneously handle data from different sources, e.g. different interventional distributions. ASD-JCI123kt is a hybrid causal discovery algorithm that scores how well each hypothetical causal graph matches the (strengths of the) observed dependences in the pooled data, giving more weight to stronger dependences. As an independence test, we again make use of the $G^2$-test for conditional independence of discrete variables with $p$-value threshold $\alpha = 0.025$ (we do not use ASD-JCI123kt with the linear Gaussian experiment). Contrary to the direct testing baseline, ASD-JCI123kt combines all conditional independence test results in order to score the underlying causal graph(s). Since the algorithm makes use of an Answer Set Program (ASP) building on work by Hyttinen et al. (2014), it is straightforward to query the ASP optimizer for separating sets (e.g., how much evidence is there that variable $V_i$ is independent of $V_j$ given $\mathbf{S}$), by applying the feature scoring approach proposed by Magliacane et al. (2016). We accept a set $\mathbf{S}$ as a valid separating set if for all $I \in \mathbf{I}$, the confidence score for the independence $I \perp\!\!\!\perp_{\mathbb{P}_{\mathcal{M}}} Y \,|\, \mathbf{S}$ output by ASD-JCI123kt is positive. This causal discovery algorithm is of particular interest because it tries to find a graph that matches all independencies simultaneously. Therefore, while its running time is particularly slow, it might show behaviour where it corrects direct independence test results if these contradict other observed independencies, which may lead to improved performance compared to direct testing. On the contrary, faster causal discovery algorithms which achieve this speed by pruning the large space of graphs generally do not override the results of any of the conditional independence tests they perform.

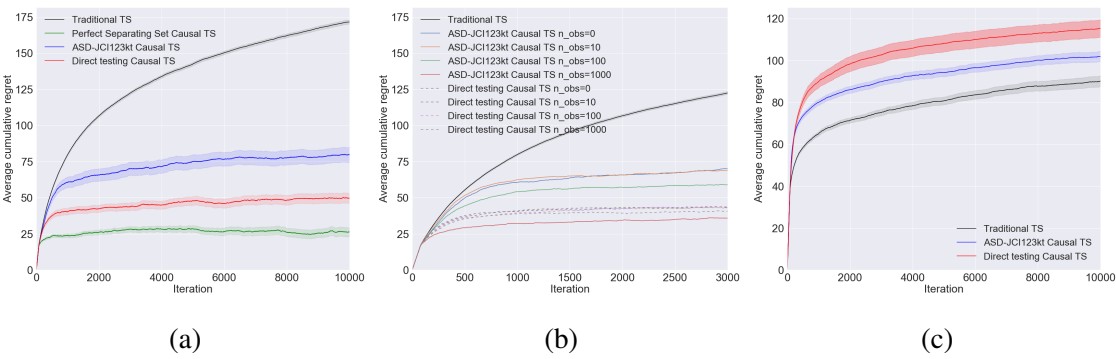

(a) (b) (c)

Figure 2: (a): Simulation results on all 234 DAGs of 4 nodes where $Y$ has at least one parent. (b): Simulation results of direct testing variant on graphs of $4, 5$ and $6$ nodes. (c): Simulation result on baseline tests where no separating set exists.

## 5. Experimentation

We now proceed to simulate several Causal Bandit problems. We leave hyperparameter tuning as a further optimization challenge for the future, setting the hyperparameters as discussed in the causal discovery methods section. For Monte Carlo estimation in the Causal Thompson Sampling and bootstrapping in Causal UCB-Normal we use 200 and 50 samples respectively. We compare our causal Thompson Sampling variant to a naïve Thompson Sampling baseline, and our causal UCB normal variant to a naïve UCB normal baseline. Further details on the simulation studies are provided in the appendix.

### 5.1. Simulation studies for Causal Thompson Sampling

Unfortunately, in terms of scale we are limited to small graphs, due to the running time of causal discovery (this is especially the case for ASD-JCI123kt, though direct testing is also slow due to testing every possible subset of variables). As a first experiment, we generated all acyclic causal graphs $G = (\mathbf{V}, \mathcal{E})$ over $4$ binary variables with no confounders and compare the cumulative regret. We allow perfect interventions on all subsets of variables excluding the target variable, thus there are $3^3 = 27$ possible actions. We only generate graphs where $Y$ has at least $1$ parent (otherwise the regret is always $0$). Permutations of the variables excluding $Y$ are disregarded. Results are plotted in Figure 2(a), where we also show the performance of a variant of Causal Thompson Sampling that has access to a perfect separating set as prior knowledge, where as separating set we choose the parents of $Y$. We see that our causal TS algorithm significantly outperforms classical Thompson Sampling. Interestingly, direct testing outperforms full causal discovery in this experiment. With our choice of hyperparameters, ASD-JCI123kt seems more conservative than direct testing, i.e. it is less sensitive but also has less false positives (see Figure 3(b)). This seems to be to its detriment in this experiment, where most graphs that are generated have many separating sets. Obviously, having prior knowledge of a perfect separating set yields even better performance.

Secondly, to investigate how our method scales with more nodes, we randomly generate graphs of $4, 5$ and $6$ variables, and again allow interventions on any subset of nodes excluding $Y$. The random graph generation scheme is such that $Y$ is always the last node in the topological order and such that there is always at least $1$ separating set. To keep running time reasonable, we only use direct

testing in this experiment. Results are shown in Figure 2(b). It is clear that the advantage of Causal TS over classical TS improves substantially with more variables. This is expected, as the number of arms exponentially increases with the number of variables, thus allowing more data sharing and a richer signal for independence tests.

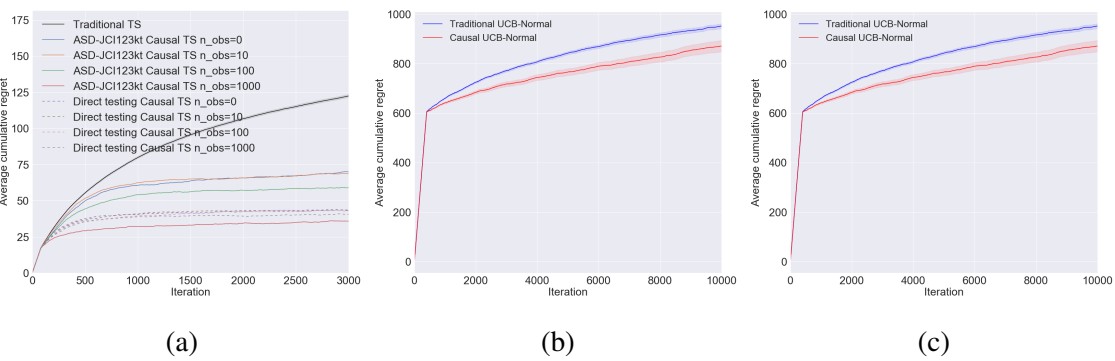

(a)                                         (b)                                         (c)

Figure 3: (a): Simulation results on all 234 DAGs of 4 nodes where the algorithms are given $n_{obs}$ additional observational samples before the start of the bandit game. (b): False positive rate and sensitivity by number of iterations in the simulation study with all DAGs of 4 nodes. (c): Results of Causal Bandit experiment based on Sachs data.

We also run a baseline test where we try to confuse our independence tests and causal discovery. Data for the variables are generated by exactly the same process as for our all 4 node graphs test. However, after we generate our data, we throw away the value generated for $Y$, and replace it with a Bernoulli random variable where the mean is randomly chosen and is different for each possible setting of the intervention variables (i.e. it is a classical bandit and there is no viable separating set for any of the possible interventions). The results are in Figure 2(c). In this case where there are no separating sets, the penalty for still trying to find and exploit them is comparatively low. We also see that the conservativeness of ASD-JCI123kt is to its benefit compared to direct testing in this experiment.

Our framework is distinct from previous approaches in that it does not assume perfectly known distributions a-priori, but may directly benefit from finite observational (or interventional) data that is available before the experiment starts. To investigate this, we construct an experiment where we give the causal algorithms a number of observational samples beforehand. This experiment is particularly interesting for ASD-JCI123kt, since we see how purely observational samples may help to correct conditional independence test errors that occur when directly testing for a separating set. Results are shown in figure 3(a). It is clear that both direct testing and ASD-JCI123kt variants benefit from additional observational data. However, ASD seems to benefit significantly more, as it seems to be able to combine data of the different interventional regimes with the observational regime effectively to improve its causal discovery performance. This result makes a case for causal discovery algorithms that have an error-correcting property.

## 5.2. Sachs experiment for Causal UCB-Normal

Finally, we experiment with a Causal Bandit where data is generated by sampling from a popular real-world dataset by Sachs et al. (2005). In this experiment, human immune cells are randomly

assigned to batches. Interventions are then performed on each batch separately, by adding some chemical compound to each batch. Then, properties of the each cell within a batch are measured. As is commonly done, we ignore experiments with ICAM-2. The other experiments are encoded into intervention variables (as is described in Mooij et al. (2020) in more elaborate detail). This results in 8 possible actions, corresponding to 8 possible settings of the 6 context variables, including the "observational" intervention). As endogenous variables, to limit computation time, we select Mek, Erk, Akt and PKC. As target variable we choose Raf. These variables were selected after studying the estimated causal graphs in Mooij et al. (2020), where we tried to select a set of variables which may actually be likely to have separating sets. This is an important step: for a random choice of variables the data from the Sachs experiment seems too confounded for there to be an efficient separating set, thus resulting in a slight reduction in performance compared to the baseline (see appendix D). We only run direct testing on this experiment in order to keep computation time reasonable. We turn this into a Causal Bandit by randomly selecting an appropiate sample from the dataset each time an action is selected. We let each algorithm sample 50 uniformly selected actions at the start of the experiment to improve stability of independence tests.

Results of this experiment are shown in figure 3(c). We see that in this case Causal Normal UCB outperforms traditional Normal UCB. This is especially interesting when considering that when we test for conditional independence between context variables and the target variable on the *full* dataset, we do not find any with the chosen treshold. Despite this, Causal Normal UCB seems to provide an inductive bias with a favorable bias-variance tradeoff.

## 6. Conclusion

We have shown that exploiting separating sets in Causal Bandit problems may yield significantly improved performance compared to traditional bandit algorithms, even when there is no prior knowledge of the graph or of interventional distributions. We employed causal discovery algorithms and direct testing to estimate separating sets from the data in an online fashion, which we then leveraged using our information sharing estimator. In our experiments, we found that in cases where there is structure that can be exploited, our algorithms show significantly improved performance compared to naïve bandit algorithms.

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

## Appendix A. Proof of theorems

In this appendix section, we set out to prove the theorem stated in the main paper. Here it is convenient to introduce vectorized notation for the relevant quantities. Let us consider our estimator given a particular separating set $\mathbf{S}$ with domain $D(\mathbf{S})$. We define the following vectors indexed by $D(\mathbf{S})$, such that the value at index $\mathbf{s} \in \mathbf{S}$ is given by:

$$\left(\hat{\mathbf{p}}_{\mathbf{S}}(\cdot \mid \boldsymbol{\zeta}; \mathcal{D}^N)\right)_{\mathbf{s}} = \hat{p}(\mathbf{s} \mid \boldsymbol{\zeta}, \mathcal{D}^N), \tag{7}$$

$$(\mathbf{p}_{\mathbf{S}}(\cdot \mid \boldsymbol{\zeta}))_{\mathbf{s}} = \mathbb{P}[\mathbf{S} = \mathbf{s} \mid \mathbf{I} = \boldsymbol{\zeta}], \tag{8}$$

$$\left(\hat{\boldsymbol{\mu}}_{\mathbf{S}}(y \mid \mathcal{D}^N)\right)_{\mathbf{s}} = \hat{\mu}(y \mid \mathbf{s}; \mathcal{D}^N), \tag{9}$$

$$(\boldsymbol{\mu}_{\mathbf{S}})_{\mathbf{s}} = \mathbb{E}[Y = 1 \mid \mathbf{S} = \mathbf{s}], \tag{10}$$

$$\left(\mathbf{N}_{\mathbf{S}, \mathcal{D}^N}(p)\right)_{\mathbf{s}} = \mathcal{N}_{\mathcal{D}^N}(\mathbf{S} = \mathbf{s} \wedge p). \tag{11}$$

With this in hand, we can write the definition of our information sharing estimator (2) as an inner product:

$$\hat{\mu}_{IS}(y \mid \boldsymbol{\zeta}; \mathcal{D}^N, \mathbf{S}) = \hat{\mathbf{p}}_{\mathbf{S}}^{\mathsf{T}}(\cdot \mid \boldsymbol{\zeta}; \mathcal{D}^N) \hat{\boldsymbol{\mu}}_{\mathbf{S}}(y \mid \mathcal{D}^N). \tag{12}$$

One minor detail is that it is possible that for a certain value of $\mathbf{s} \in D(\mathbf{S})$ we have no samples, and thus $\hat{\mu}(y \mid \mathbf{s}; \mathcal{D}^N)$ is undefined. But then for that value $\hat{p}(\mathbf{s} \mid \boldsymbol{\zeta}, \mathcal{D}^N)$ must also be 0 (otherwise we would have at least one sample), and we consider the product to be 0. Since it will be multiplied by 0 regardless, in the analysis that follows we may consider the expectation of $\hat{\mu}(y \mid \mathbf{s}; \mathcal{D}^N)$ to be the same as if we had at least one sample, i.e. the true mean $\mathbb{E}[Y = 1 \mid \mathbf{S} = \mathbf{s}]$.

### A.1. Proof of Theorem 3.1

We set out to prove the theorem:

**Theorem 3.1** *If we calculate $\hat{\boldsymbol{\mu}}_{\mathbf{S}}(y \mid \mathcal{D}^N)$ from a dataset $\mathcal{D}^N$ where we have a fixed number of i.i.d. data points from each possible intervention generated by a discrete Causal Bandit, and $\mathbf{I} \perp\!\!\!\perp_{\mathbb{P}_{\mathcal{M}}} Y \mid \mathbf{S}$, and there is at least one sample from intervention $\mathbf{I} = \boldsymbol{\zeta}$, then the information sharing estimator (2) is unbiased. Furthermore, its variance is upper bounded by that of the sample mean:*

$$\mathbb{V}[\hat{\mu}_{IS}(y \mid \boldsymbol{\zeta}; \mathcal{D}^N, \mathbf{S})] \leq \mathbb{V}[\hat{\mu}_{SM}(y \mid \boldsymbol{\zeta}; \mathcal{D}^N)] \tag{13}$$

We consider the information sharing estimator (12) calculated from data generated by a bandit environment, where we have a certain fixed number of i.i.d. datapoints from each possible setting of $\mathbf{I}$. We assume we have at least one sample from the intervention of interest $\boldsymbol{\zeta}$. We first show that

the vectors in (12) are uncorrelated, which has as immediate corollary that the information sharing estimator is unbiased. This follows from the law of total expectation:

$$
\begin{aligned}
\mathbb{E}\left[\hat{\mathbf{p}}_{\mathbf{S}}^{\mathsf{T}}(\cdot\mid\boldsymbol{\zeta};\mathcal{D}^N)\hat{\boldsymbol{\mu}}_{\mathbf{S}}(y\mid\mathcal{D}^N)\right] &= \mathbb{E}\left[\mathbb{E}\left[\hat{\mathbf{p}}_{\mathbf{S}}^{\mathsf{T}}(\cdot\mid\boldsymbol{\zeta};\mathcal{D}^N)\hat{\boldsymbol{\mu}}_{\mathbf{S}}(y\mid\mathcal{D}^N)\mid\mathbf{N}_{\mathbf{S},\mathcal{D}^N}(\mathbf{I}=\boldsymbol{\zeta}),\mathbf{N}_{\mathbf{S},\mathcal{D}^N}(\mathbf{I}\neq\boldsymbol{\zeta})\right]\right] \\
&= \mathbb{E}\left[\hat{\mathbf{p}}_{\mathbf{S}}^{\mathsf{T}}(\cdot\mid\boldsymbol{\zeta};\mathcal{D}^N)\mathbb{E}\left[\hat{\boldsymbol{\mu}}_{\mathbf{S}}(y\mid\mathcal{D}^N)\mid\mathbf{N}_{\mathbf{S},\mathcal{D}^N}(\mathbf{I}=\boldsymbol{\zeta}),\mathbf{N}_{\mathbf{S},\mathcal{D}^N}(\mathbf{I}\neq\boldsymbol{\zeta})\right]\right] \\
&= \mathbb{E}\left[\hat{\mathbf{p}}_{\mathbf{S}}^{\mathsf{T}}(\cdot\mid\boldsymbol{\zeta};\mathcal{D}^N)\right]\boldsymbol{\mu}_{\mathbf{S}} \\
&= \mathbf{p}_{\mathbf{S}}^{\mathsf{T}}(\boldsymbol{\zeta}|\mathcal{D}^N)\boldsymbol{\mu}_{\mathbf{S}} = \mathbb{E}[Y\mid\mathbf{I}=\boldsymbol{\zeta}]
\end{aligned}
$$

where in the second line we use that $\hat{\mathbf{p}}_{\mathbf{S}}(\cdot\mid\boldsymbol{\zeta};\mathcal{D}^N)$ is deterministic conditional on $\mathbf{N}_{\mathbf{S},\mathcal{D}^N}(\mathbf{I}=\boldsymbol{\zeta})$ and thus it factors out of the inner expectation. On the third line, we use that conditionally on the counts $\mathcal{N}_{\mathcal{D}^N}(\mathbf{S}=\mathbf{s})$ (which is a deterministic function of the vectors we condition on), $\hat{\mu}(\mathbf{s}\mid\mathcal{D}^N)$ is just the mean of $\mathcal{N}_{\mathcal{D}^N}(\mathbf{S}=\mathbf{s})$ Bernoulli variables and thus unbiased, and thus the inner expectation evaluates to the vector of true means $\boldsymbol{\mu}_{\mathbf{S}}$ and factors out. The exact same conditioning argument using the law of total expectation can be used to show that $\mathbb{E}\left[\hat{\boldsymbol{\mu}}_{\mathbf{S}}(y\mid\mathcal{D}^N)\right] = \boldsymbol{\mu}_{\mathbf{S}}$, from which it follows that the expectation factors and thus the vectors are uncorrelated. Finally in the fourth line, the elements of $\hat{\mathbf{p}}_{\mathbf{S}}(\cdot\mid\boldsymbol{\zeta};\mathcal{D}^N)$ can be seen as the mean of at least one Bernoulli variable (by assumption) and thus are unbiased, from which the unbiasedness of the information sharing estimator follows.

We analyze the variance using a similar strategy, using the law of total variance, adding the same conditioning we did to show unbiasedness:

$$
\begin{aligned}
\mathbb{V}\left[\hat{\mu}_{IS}(y\mid\boldsymbol{\zeta};\mathcal{D}^N,\mathbf{S})\right] = {}&\mathbb{E}\left[\mathbb{V}\left[\hat{\mathbf{p}}_{\mathbf{S}}^{\mathsf{T}}(\cdot\mid\boldsymbol{\zeta};\mathcal{D}^N)\hat{\boldsymbol{\mu}}_{\mathbf{S}}(y\mid\mathcal{D}^N)\mid\mathbf{N}_{\mathbf{S},\mathcal{D}^N}(\mathbf{I}=\boldsymbol{\zeta}),\mathbf{N}_{\mathbf{S},\mathcal{D}^N}(\mathbf{I}\neq\boldsymbol{\zeta})\right]\right] \\
&+ \mathbb{V}\left[\mathbb{E}\left[\hat{\mathbf{p}}_{\mathbf{S}}^{\mathsf{T}}(\cdot\mid\boldsymbol{\zeta};\mathcal{D}^N)\hat{\boldsymbol{\mu}}_{\mathbf{S}}(y\mid\mathcal{D}^N)\mid\mathbf{N}_{\mathbf{S},\mathcal{D}^N}(\mathbf{I}=\boldsymbol{\zeta}),\mathbf{N}_{\mathbf{S},\mathcal{D}^N}(\mathbf{I}\neq\boldsymbol{\zeta})\right]\right].
\end{aligned}
$$

Now note, in both terms, $\hat{\mathbf{p}}_{\mathbf{S}}(\cdot\mid\boldsymbol{\zeta};\mathcal{D}^N)$ is non-random. In the second term this vector factors out because of linearity of expectation. In the case of the first term, the individual elements of $\hat{\boldsymbol{\mu}}_{\mathbf{S}}$ are uncorrelated with each-other since they are calculated from disjoint sets of data, thus this vector factors out of the variance element wise squared. This yields:

$$
\begin{aligned}
\mathbb{V}\left[\hat{\mu}_{IS}(y\mid\boldsymbol{\zeta};\mathcal{D}^N,\mathbf{S})\right] = {}&\mathbb{E}\left[\left(\hat{\mathbf{p}}_{\mathbf{S}}^{\mathsf{T}}(\cdot\mid\boldsymbol{\zeta};\mathcal{D}^N)\right)^2\mathbb{V}\left[\hat{\boldsymbol{\mu}}_{\mathbf{S}}(y\mid\mathcal{D}^N)\mid\mathbf{N}_{\mathbf{S},\mathcal{D}^N}(\mathbf{I}=\boldsymbol{\zeta}),\mathbf{N}_{\mathbf{S},\mathcal{D}^N}(\mathbf{I}\neq\boldsymbol{\zeta})\right]\right] \\
&\hspace{10cm}(14)\\
&+ \mathbb{V}\left[\hat{\mathbf{p}}_{\mathbf{S}}^{\mathsf{T}}(\cdot\mid\boldsymbol{\zeta};\mathcal{D}^N)\mathbb{E}\left[\hat{\boldsymbol{\mu}}_{\mathbf{S}}(y\mid\mathcal{D}^N)\mid\mathbf{N}_{\mathbf{S},\mathcal{D}^N}(\mathbf{I}=\boldsymbol{\zeta}),\mathbf{N}_{\mathbf{S},\mathcal{D}^N}(\mathbf{I}\neq\boldsymbol{\zeta})\right]\right]
\end{aligned}
$$

where the square of the vector in the first term is elementwise, and the variance of a vector in the first term is just the vector of diagonal elements of the covariance matrix, i.e. there are no covariance terms. In the second term, we may now again use that the inner expectation is unbiased following the same argument as before. For the first term, the variance vector $\mathbb{V}\left[\hat{\boldsymbol{\mu}}_{\mathbf{S}}(y\mid\mathcal{D}^N)\mid\mathbf{N}_{\mathbf{S},\mathcal{D}^N}(\mathbf{I}=\boldsymbol{\zeta}),\mathbf{N}_{\mathbf{S},\mathcal{D}^N}(\mathbf{I}\neq\boldsymbol{\zeta})\right] = \boldsymbol{\mu}_S\otimes(1-\boldsymbol{\mu}_S)\oslash\mathbf{N}_{\mathbf{S},\mathcal{D}^N}(\top)$ is also well defined as the variance of a sample mean of a set of Bernoulli random variables, where $\otimes$ is elementwise product, $\oslash$ is elementwise division and $\mathbf{N}_{\mathbf{S},\mathcal{D}^N}(\top) = \mathbf{N}_{\mathbf{S},\mathcal{D}^N}(\mathbf{I}=\boldsymbol{\zeta})+\mathbf{N}_{\mathbf{S},\mathcal{D}^N}(\mathbf{I}\neq\boldsymbol{\zeta})$. Substituting

this into (14) yields:

$$\mathbb{V}\left[\hat{\mu}_{IS}(y\,|\,\boldsymbol{\zeta};\mathcal{D}^N,\mathbf{S})\right] = \mathbb{E}\left[\left(\hat{\mathbf{p}}_{\mathbf{S}}(\cdot\,|\,\boldsymbol{\zeta};\mathcal{D}^N)\right)^2 \oslash \mathbf{N}_{\mathbf{S},\mathcal{D}^N}(\top)\right]^{\mathsf{T}} \boldsymbol{\mu}_{\mathbf{S}} \otimes (\mathbf{1}-\boldsymbol{\mu}_{\mathbf{S}}) \qquad (15)$$
$$+ \mathbb{V}\left[\hat{\mathbf{p}}_{\mathbf{S}}(\cdot\,|\,\boldsymbol{\zeta};\mathcal{D}^N)\boldsymbol{\mu}_{\mathbf{S}}\right]$$

Interestingly, the second term corresponds to our information leakage estimator if we were given perfect oracle estimates $\boldsymbol{\mu}_{\mathbf{S}}$. Since the advantage gained by the information sharing estimator is through better estimation of $\boldsymbol{\mu}_{\mathbf{S}}$, this term can be seen as a base error that cannot be reduced through information leakage.

We evaluate the second term. Let $\mathbf{s}_1,\ldots,\mathbf{s}_{\mathcal{N}_{\mathcal{D}^N}(\mathbf{I}=\boldsymbol{\zeta})}$ be the one-hot vector encoded values of $\mathbf{S}$ observed in the subset of our data where $\mathbf{I}=\boldsymbol{\zeta}$. These are i.i.d. categorical variables, and since $\hat{\mathbf{p}}_{\mathbf{S}}(\cdot\,|\,\boldsymbol{\zeta};\mathcal{D}^N) = \frac{1}{\mathcal{N}_{\mathcal{D}^N}(\mathbf{I}=\boldsymbol{\zeta})}\sum_{k=1}^{\mathcal{N}_{\mathcal{D}^N}(\mathbf{I}=\boldsymbol{\zeta})}\mathbf{s}_k$, the second term becomes by independence:

$$\mathbb{V}\left[\hat{\mathbf{p}}_{\mathbf{S}}(\cdot\,|\,\boldsymbol{\zeta};\mathcal{D}^N)\boldsymbol{\mu}_{\mathbf{S}}\right] = \mathbb{V}\left[\frac{1}{\mathcal{N}_{\mathcal{D}^N}(\mathbf{I}=\boldsymbol{\zeta})}\sum_{k=1}^{\mathcal{N}_{\mathcal{D}^N}(\mathbf{I}=\boldsymbol{\zeta})}\mathbf{s}_k^{\mathsf{T}}\boldsymbol{\mu}_{\mathbf{S}}\right]$$
$$= \frac{1}{\mathcal{N}_{\mathcal{D}^N}(\mathbf{I}=\boldsymbol{\zeta})}\mathbb{V}\left[\mathbf{s}_1^{\mathsf{T}}\boldsymbol{\mu}_{\mathbf{S}}\right]$$
$$= \frac{1}{\mathcal{N}_{\mathcal{D}^N}(\mathbf{I}=\boldsymbol{\zeta})}\mathbb{V}_{\mathbf{s}\sim\mathbb{P}[\mathbf{S}=\mathbf{s}|\mathbf{I}=\boldsymbol{\zeta}]}\left[\mathbb{E}[Y|\mathbf{S}=\mathbf{s}]\right]. \qquad (16)$$

Let us now turn our attention to the first term of equation (15). This is an inner product between vectors, where the left factor is an expectation of a vector. Let us consider an element of this expectation vector at index $\mathbf{s}\in D(\mathbf{S})$:

$$\left(\mathbb{E}\left[\left(\hat{\mathbf{p}}_{\mathbf{S}}(\cdot\,|\,\boldsymbol{\zeta};\mathcal{D}^N)\right)^2\oslash\mathbf{N}_{\mathbf{S},\mathcal{D}^N}(\top)\right]\right)_{\mathbf{s}} = \mathbb{E}\left[\frac{\hat{p}^2(\mathbf{s}\,|\,\boldsymbol{\zeta},\mathcal{D}^N)}{\mathcal{N}_{\mathcal{D}^N}(\mathbf{S}=\mathbf{s})}\right]$$
$$= \mathbb{E}\left[\frac{\hat{p}(\mathbf{s}\,|\,\boldsymbol{\zeta},\mathcal{D}^N)}{\mathcal{N}_{\mathcal{D}^N}(\mathbf{I}=\boldsymbol{\zeta})}\frac{\mathcal{N}_{\mathcal{D}^N}(\mathbf{S}=\mathbf{s},\mathbf{I}=\boldsymbol{\zeta})}{\mathcal{N}_{\mathcal{D}^N}(\mathbf{S}=\mathbf{s})}\right]$$
$$= \mathbb{E}\left[\frac{\hat{p}(\mathbf{s}\,|\,\boldsymbol{\zeta},\mathcal{D}^N)}{\mathcal{N}_{\mathcal{D}^N}(\mathbf{I}=\boldsymbol{\zeta})}\left(1-\frac{\mathcal{N}_{\mathcal{D}^N}(\mathbf{S}=\mathbf{s},\mathbf{I}\neq\boldsymbol{\zeta})}{\mathcal{N}_{\mathcal{D}^N}(\mathbf{S}=\mathbf{s})}\right)\right]$$
$$= \frac{1}{\mathcal{N}_{\mathcal{D}^N}(\mathbf{I}=\boldsymbol{\zeta})}\mathbb{E}\left[\hat{p}(\mathbf{s}\,|\,\boldsymbol{\zeta},\mathcal{D}^N)(1-\alpha(\mathbf{s},\boldsymbol{\zeta},\mathcal{D}^N))\right] \quad (17)$$

where:

$$\alpha(\mathbf{s},\boldsymbol{\zeta},\mathcal{D}^N) := \frac{\mathcal{N}_{\mathcal{D}^N}(\mathbf{S}=\mathbf{s},\mathbf{I}\neq\boldsymbol{\zeta})}{\mathcal{N}_{\mathcal{D}^N}(\mathbf{S}=\mathbf{s})}. \qquad (18)$$

Note that $\alpha(\mathbf{s},\boldsymbol{\zeta},\mathcal{D}^N)$ equals 0 if $\mathcal{N}_{\mathcal{D}^N}(\mathbf{S}=\mathbf{s},\mathbf{I}\neq\boldsymbol{\zeta})=0$ (i.e. there is no additional data to use where $\mathbf{I}\neq\boldsymbol{\zeta}$ for information sharing for this value of $\mathbf{s}$), and approaches 1 if $\mathcal{N}_{\mathcal{D}^N}(\mathbf{S}=\mathbf{s},\mathbf{I}\neq\boldsymbol{\zeta})$ goes to $\infty$ and we keep $\mathcal{N}_{\mathcal{D}^N}(\mathbf{S}=\mathbf{s},\mathbf{I}=\boldsymbol{\zeta})$ fixed, since in the denominator $\mathcal{N}_{\mathcal{D}^N}(\mathbf{S}=\mathbf{s}) = \mathcal{N}_{\mathcal{D}^N}(\mathbf{S}=\mathbf{s},\mathbf{I}=\boldsymbol{\zeta})+\mathcal{N}_{\mathcal{D}^N}(\mathbf{S}=\mathbf{s},\mathbf{I}\neq\boldsymbol{\zeta})$.

Substituting (16) and (17) into (15), the variance of the information sharing estimator then becomes

$$\mathbb{V}\left[\hat{\mu}_{IS}(y\,|\,\boldsymbol{\zeta};\mathcal{D}^N,\mathbf{S})\right] = \frac{1}{\mathcal{N}_{\mathcal{D}^N}(\mathbf{I}=\boldsymbol{\zeta})}\left(\mathbb{E}\left[\hat{\mathbf{p}}_{\mathbf{S}}(\cdot\,|\,\boldsymbol{\zeta};\mathcal{D}^N)\otimes(\mathbf{1}-\boldsymbol{\alpha}(\boldsymbol{\zeta},\mathcal{D}^N)))\right]^{\mathsf{T}}\boldsymbol{\mu}_S\otimes(\mathbf{1}-\boldsymbol{\mu}_{\mathbf{S}})\right.$$
$$\left.+\,\mathbb{V}_{\mathbf{s}\sim\mathbb{P}[\mathbf{S}=\mathbf{s}|\mathbf{I}=\boldsymbol{\zeta}]}\left[\mathbb{E}\left[Y|\mathbf{S}=\mathbf{s}\right]\right]\right) \tag{19}$$

where we define $\boldsymbol{\alpha}(\boldsymbol{\zeta},\mathcal{D}^N)$ as the vectorized version of $\alpha(\mathbf{s},\boldsymbol{\zeta},\mathcal{D}^N)$ indexed by $\mathbf{s}\in D(\mathbf{S})$ such that $\left(\boldsymbol{\alpha}(\boldsymbol{\zeta},\mathcal{D}^N)\right)_{\mathbf{s}}=\alpha(\mathbf{s},\boldsymbol{\zeta},\mathcal{D}^N)$. Let us now consider the term which has $\boldsymbol{\alpha}(\boldsymbol{\zeta},\mathcal{D}^N)$ as a factor when we expand the parenthesis inside the expectation of the first term. From its definition, we see that $\boldsymbol{\alpha}(\boldsymbol{\zeta},\mathcal{D}^N)$ is elementwise upper bounded by $\mathbf{1}$ (at infinitely many samples where $\mathbf{I}\neq\boldsymbol{\zeta}$ and a finite number of samples $\mathbf{I}=\boldsymbol{\zeta}$) for all values of $\mathbf{s}$), and elementwise lower bounded by $\mathbf{0}$ if we have no samples where $\mathbf{I}=\boldsymbol{\zeta}$. Therefore, since all values are positive, if we define:

$$\alpha^*(\boldsymbol{\zeta},\mathcal{D}^N) = \frac{\mathbb{E}[\hat{\mathbf{p}}_{\mathbf{S}}(\cdot\,|\,\boldsymbol{\zeta};\mathcal{D}^N)\otimes\boldsymbol{\alpha}(\boldsymbol{\zeta},\mathcal{D}^N)]^{\mathsf{T}}\boldsymbol{\mu}_{\mathbf{S}}\otimes(\mathbf{1}-\boldsymbol{\mu}_S)}{\mathbb{E}[\hat{\mathbf{p}}_{\mathbf{S}}(\cdot\,|\,\boldsymbol{\zeta};\mathcal{D}^N)\otimes\mathbf{1}]^{\mathsf{T}}\boldsymbol{\mu}_{\mathbf{S}}\otimes(\mathbf{1}-\boldsymbol{\mu}_S)} \tag{20}$$

then $\alpha^*(\boldsymbol{\zeta},\mathcal{D}^N)$ is upper bounded by $1$ since from its definition we see that $\boldsymbol{\alpha}(\boldsymbol{\zeta},\mathcal{D}^N)$ is elementwise upper bounded by $\mathbf{1}$ in which case the numerator and denominator are equal. Furthermore, $\alpha^*(\boldsymbol{\zeta},\mathcal{D}^N)$ is lower bounded by $0$ since all values are nonnegative. Then $\alpha^*(\boldsymbol{\zeta},\mathcal{D}^N)\in[0,1)$ and substitution of $\alpha^*(\boldsymbol{\zeta},\mathcal{D}^N)$ into (19) yields:

$$\mathbb{V}\left[\hat{\mu}_{IS}(y\,|\,\boldsymbol{\zeta};\mathcal{D}^N,\mathbf{S})\right] = \frac{1}{\mathcal{N}_{\mathcal{D}^N}(\mathbf{I}=\boldsymbol{\zeta})}\left((1-\alpha^*(\boldsymbol{\zeta},\mathcal{D}^N))\mathbb{E}[\hat{\mathbf{p}}_{\mathbf{S}}(\cdot\,|\,\boldsymbol{\zeta};\mathcal{D}^N)]^{\mathsf{T}}\boldsymbol{\mu}_S\otimes(\mathbf{1}-\boldsymbol{\mu}_{\mathbf{S}})\right.$$
$$\left.+\,\mathbb{V}_{\mathbf{s}\sim\mathbb{P}[\mathbf{S}=\mathbf{s}|\mathbf{I}=\boldsymbol{\zeta}]}[\mathbb{E}[Y|\mathbf{S}=\mathbf{s}]]\right)$$
$$= \frac{1}{\mathcal{N}_{\mathcal{D}^N}(\mathbf{I}=\boldsymbol{\zeta})}\left((1-\alpha^*(\boldsymbol{\zeta},\mathcal{D}^N))\mathbf{p}_{\mathbf{S}}^{\mathsf{T}}(\boldsymbol{\zeta})\boldsymbol{\mu}_S\otimes(\mathbf{1}-\boldsymbol{\mu}_{\mathbf{S}})\right.$$
$$\left.+\,\mathbb{V}_{\mathbf{s}\sim\mathbb{P}[\mathbf{S}=\mathbf{s}|\mathbf{I}=\boldsymbol{\zeta}]}[\mathbb{E}[Y|\mathbf{S}=\mathbf{s}]]\right)$$
$$= \frac{1}{\mathcal{N}_{\mathcal{D}^N}(\mathbf{I}=\boldsymbol{\zeta})}\left(\mathbb{V}_{\mathbf{s}\sim\mathbb{P}[\mathbf{S}=\mathbf{s}|\mathbf{I}=\boldsymbol{\zeta}]}[\mathbb{E}[Y|\mathbf{S}=\mathbf{s}]]\right.$$
$$\left.+\,(1-\alpha^*(\boldsymbol{\zeta},\mathcal{D}^N))\mathbb{E}_{\mathbf{s}\sim\mathbb{P}[\mathbf{S}=\mathbf{s}|\mathbf{I}=\boldsymbol{\zeta}]}[\mathbb{E}[Y|\mathbf{S}=\mathbf{s}](1-\mathbb{E}[Y|\mathbf{S}=\mathbf{s}])]\right). \tag{21}$$

From here it is easy to see that the theorem holds, since $\alpha^*(\boldsymbol{\zeta},\mathcal{D}^N)\in[0,1)$ and the expectation in the second term is always nonnegative. Thus we may obtain an upper bound by setting $\alpha^*(\boldsymbol{\zeta},\mathcal{D}^N)=0$. Then we can combine the variance and expectation into 1 expectation, and we obtain:

$$\mathbb{V}\left[\hat{\mu}_{IS}(y\,|\,\boldsymbol{\zeta};\mathcal{D}^N,\mathbf{S})\right] \leq \frac{1}{\mathcal{N}_{\mathcal{D}^N}(\mathbf{I}=\boldsymbol{\zeta})}\left(\mathbb{V}_{\mathbf{s}\sim\mathbb{P}[\mathbf{S}=\mathbf{s}|\mathbf{I}=\boldsymbol{\zeta}]}[\mathbb{E}[Y|\mathbf{S}=\mathbf{s}]]\right.$$
$$\left.+\,\mathbb{E}_{\mathbf{s}\sim\mathbb{P}[\mathbf{S}=\mathbf{s}|\mathbf{I}=\boldsymbol{\zeta}]}[\mathbb{E}[Y|\mathbf{S}=\mathbf{s}](1-\mathbb{E}[Y|\mathbf{S}=\mathbf{s}])]\right),$$
$$= \frac{1}{\mathcal{N}_{\mathcal{D}^N}(\mathbf{I}=\boldsymbol{\zeta})}\mathbb{V}[Y|\mathbf{I}=\boldsymbol{\zeta}] = \mathbb{V}[\mu_{SM}(\boldsymbol{\zeta};\mathcal{D}^N)], \tag{22}$$

by law of total variance, which is what was to be shown.

The value of $\alpha^*(\zeta, \mathcal{D}^N)$ is a complicated inner product depending on the model parameters, and is a measure of the expected relative sizes of $\mathcal{N}_{\mathcal{D}^N}(\mathbf{S} = \mathbf{s}|\mathbf{I} = \zeta)$ and $\mathcal{N}_{\mathcal{D}^N}(\mathbf{S} = \mathbf{s}|\mathbf{I} \neq \zeta)$ for the values of $\mathbf{s}$ where $\mathbb{P}[\mathbf{S} = \mathbf{s}|\mathbf{I} = \zeta]$ is large.

It is easy to see that $\alpha^*(\zeta, \mathcal{D}^N) \geq \min_{\mathbf{s}} \alpha(\mathbf{s}, \zeta)$, since then $\boldsymbol{\alpha}(\zeta, \mathcal{D}^N) \geq \min_{\mathbf{s}} \alpha(\mathbf{s}, \zeta)\mathbf{1}$ elementwise and we may then factor $\alpha^*(\zeta, \mathcal{D}^N)$ out of the expectation in the numerator of (20) after which the fraction cancels. An interesting case is if we condition on knowing $\{\mathcal{N}_{\mathcal{D}^N}(\mathbf{S} = \mathbf{s}, \mathbf{I} = \zeta)\}_{\mathbf{s} \in D(\mathbf{S})}$. Let us define $c$ to be the largest real number such that for all $\mathbf{s} \in D(\mathbf{S})$, it holds that $\mathcal{N}_{\mathcal{D}^N}(\mathbf{S} = \mathbf{s}, \mathbf{I} \neq \zeta) \geq c\mathcal{N}_{\mathcal{D}^N}(\mathbf{S} = \mathbf{s}, \mathbf{I} = \zeta)$. From its definition (18), we see that then $\alpha(\mathbf{s}, \zeta, \mathcal{D}^N) \geq \frac{c}{c+1}$, and thus $\alpha^*(\zeta, \mathcal{D}^N) \geq \frac{c}{c+1}$.

The relative sizes of $\mathbb{V}_{\mathbf{s} \sim \mathbb{P}[\mathbf{S}=\mathbf{s}|\mathbf{I}=\zeta]}\big[\mathbb{E}[Y|\mathbf{S} = \mathbf{s}]\big]$ and $\mathbb{E}_{\mathbf{s} \sim \mathbb{P}[\mathbf{S}=\mathbf{s}|\mathbf{I}=\zeta]}\big[\mathbb{V}[Y|\mathbf{S} = \mathbf{s}]\big]$ signify how well additional data from $\mathbf{I} \neq \zeta$ helps in estimating $\mathbb{E}[Y|\mathbf{I} = \zeta]$. Interestingly, not always beneficial to share data through information leakage. Specifically, if $\mathbb{V}[Y|\mathbf{S} = \mathbf{s}] = 0$ for all $\mathbf{s}$ in the support of $\mathbb{P}[\mathbf{S} = \mathbf{s}|\mathbf{I} = \zeta]$, then there is no error due to misestimation of $\mathbb{E}[Y|\mathbf{S} = \mathbf{s}]$ (since they are then deterministic thus if we have just 1 sample this is enough) and all error of the information sharing estimator stems from misestimation of $\mathbb{P}[\mathbf{S} = \mathbf{s}|\mathbf{I} = \zeta]$. Then no amount of additional data with $\mathbf{I} \neq \zeta$ may help. This was the case in our example for why $S_3$ is not always optimal. On the other hand, if the distributions $\mathbb{P}[\mathbf{S}|\mathbf{I} = \zeta]$ are disjoint for all different $\zeta$, then we see from equation 18 that $\boldsymbol{\alpha}$ is $\mathbf{0}$, and thus again information sharing provides no benefit. This was the case for the choice $S_1$ in our example.

## Appendix B. Additional details of experiments

In this section we provide some further details on how we set up the different simulation experiments.

### B.1. All DAG experiment

From the description in the main text it is clear what DAGs are considered. Here we specify the distribution of the discrete variables. If a variable $X$ has no parents, we define $P[X = 1] = 0.5$. For each variable with parents, for each of its parents we randomly select a "target value", either $1$ or $0$. When we generate the value of variable $X$, we count the number of parents that match their target value, let us call this $m$. Then:

$$\mathbb{P}[X = 1|Pa(X)] = \frac{1 + m}{2 + |Pa(X)|}$$

### B.2. Generating larger DAGs

For the experiment where we investigate scaling, the distribution of the endogenous variables is defined exactly the same as for the All DAG experiment. The DAGs are generated in the following way:

We first fix a topological order of the variables, and set $Y$ to be the last in the topological ordering. Starting from the last node in the topological order working to the first, for each variable $X$ except for $Y$, we then first uniformly randomly pick the maximum fan-out $f$ of this variable between 1 and the number of nodes after the variable in the ordering. Then, for $f$ times we uniformly randomly select a variable $X'$ after $X$ in the ordering, and if there is no edge yet from $X$ to $X'$, we add this

---

**Algorithm 2** Causal UCB normal

---

1: **Input:** Data: $\mathcal{D}^N = \{(\boldsymbol{\zeta}^n, \boldsymbol{v}^n)\}_{n=1}^N$, set of possible interventions $D(\mathbf{I})$, target variable $Y$, number of bootstrap samples $n_B$, Separating set algorithm: $disc\_sep\_set$

2: **Output:** Next action to pick at iteration $N + 1$

3: Initialize array $index[\boldsymbol{\zeta}]$ **for** $\boldsymbol{\zeta} \in D(\mathbf{I})$

4: $\mathbf{S\_set} \leftarrow disc\_sep\_set(\mathcal{D}^N, Y, \boldsymbol{\zeta})$

5: **for all** $\boldsymbol{\zeta} \in D(\mathbf{I})$ **do**

6:      Set $index[\boldsymbol{\zeta}] = \mu_{SM}(\boldsymbol{\zeta}; \mathcal{D}^N) + 4\sqrt{\hat{\mathbb{V}}[\mu_{SM}(\boldsymbol{\zeta}; \mathcal{D}^N)]\frac{\ln N}{\mathcal{N}_{\mathcal{D}^N}(\mathbf{I}=\boldsymbol{\zeta})}}$

7:      $best\_var = \hat{\mathbb{V}}[\mu_{SM}(\boldsymbol{\zeta}; \mathcal{D}^N)]$

8:      **for all** $(\mathbf{S}, \mathbf{I}') \in \mathbf{S\_set}$ **do**

9:          $\mathbf{I}_{NS} = \mathbf{I} \setminus \mathbf{I}'$

10:          Initialize array $samples[i]$ for $i \in \{1, \ldots, n_{MC}\}$

11:          **for** $i$ in $\{1, \ldots, n_B\}$ **do**

12:              Generate bootstrapped dataset $\mathcal{D'}^N$

13:              $samples[i] = \hat{\mu}_{IS}(\boldsymbol{\zeta}; \mathcal{D'}^N, \mathbf{S}, \mathbf{I}_{NS})$

14:          **end for**

15:          **if** $\hat{\mathbb{V}}[samples] < best\_var$ **then**

16:              $index[\boldsymbol{\zeta}] = \hat{\mu}_{IS}(\boldsymbol{\zeta}; \mathcal{D}^N, \mathbf{S}, \mathbf{I}_{NS}) + 4\sqrt{\hat{\mathbb{V}}[samples]\frac{\ln N}{\mathcal{N}_{\mathcal{D}^N}(\mathbf{I}=\boldsymbol{\zeta})}}$

17:              $best\_var = \hat{\mathbb{V}}[samples]$

18:          **end if**

19:      **end for**

20: **end for**

21: **return** $\arg\max_{\boldsymbol{\zeta}} index[\boldsymbol{\zeta}]$

---

edge to the graph. If $Y$ is selected but $Y$ already has 3 parents, we do not add an edge to it and continue as prescribed.

## Appendix C. Causal UCB-Normal algorithm

The full Causal UCB-Normal algorithm is given in algorithm 2, which is based on UCB-Normal from Auer et al. (2002). Note that if we consider just line 1-5, this is the implementation of the traditional UCB-Normal algorithm, and this is what we use as a baseline. In line 9-12, we estimate the variance of the information sharing estimator using bootstrapping, where we bootstrap by sampling with replacement in such a way that the number of samples per intervention stays the same. On line 15 we then use exactly the same index as UCB normal, where we just use the information sharing estimator (using partial separating sets as described in section 3.3) instead of the sample mean, and the sample variance of the bootstraps instead of the estimated variance of the sample mean.

## Appendix D. Sachs experiment with randomized selection of endogenous variables

When we look at the estimated causal graphs in Mooij et al. (2020), we see that the causal structure is very complicated and confounded. Therefore if we randomly select a subset of the variables as

endogenous, we can expect that no separating set exists. This is reflected in the experimental results shown in figure 4.

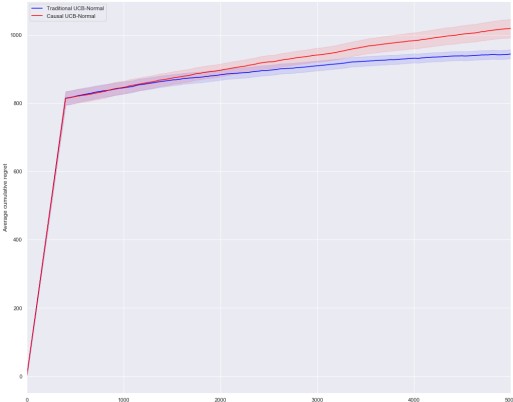

Figure 4: Simulation results on the Sachs data where we randomly select a subset of the variables as endogenous variables, 150 runs.

