# OpenReview forum: "Causal Bandits without prior knowledge using separating sets"
_cclear.cc/CLeaR/2022/Conference — CLeaR 2022 Poster_

### Official Review · Reviewer_nePs · 2021-11-20

**Confidence:** 4
**Overall Score:** 7

**Main Review:**

I have reviewed several causal bandit papers and found that some of the papers contribution is simply using the parents of reward variable Y to improve the performance. In my review, I wrote that there may be other separating set which can yield better variance reduction. Hence, I was somehow expected this idea being realized.

The authors solution is simple and clean. The use of context variable (~ regime indicator) is also a nice addition.


Some questions:
Why does the algorithms in the appendix does not check a separating set on the fly?

Do we have to perform causal discovery every iteration or once in a while (depending on p-value one may skip a few iteration)? Do we have an online version of conditional independence test or causal discovery algorithm that can be utilized in this setting?

The result with direct independence testing is better than those with causal discovery algorithm. Would it be desirable to employ some kind of “optimism in the face of uncertainty” for a “separating set”?


**Summary:**

This paper considers a causal bandit problem where a separating set between intervention and reward can be utilized through information sharing among different arms. With reduction in variance, the proposed causal bandit algorithm outperforms those without taking advantage of causal information obtainable through data and conditional independence tests.

---

> ### Author Response · Authors · 2021-12-03
> **Author response to review**
>
> Dear reviewer,
>
> Thank you for your very positive feedback! Funny coincidence that you also reviewed previous works on this precise topic. Indeed using the parents of the reward variable turns out to not be the ideal choice in general.
>
> To answer your questions:
>
> **Q:** *“Why does the algorithms in the appendix does not check a separating set on the fly?”*
>
> **A:** Both algorithms discover separating sets on line 4 by invoking disc_sep_set, where how this is done exactly is left implicit to whatever causal discovery algorithm is used. This will be better explained in section 4.
>
>  **Q:** *“Do we have to perform causal discovery every iteration or once in a while (depending on p-value one may skip a few iteration)?”*
>
> **A:** In our experiments, we run causal discovery initially, and then every time the number of data points increased by 10%. This will be made clear in the paper. (Add comment with potential of s values?)
>
> **Q:** *Do we have an online version of conditional independence test or causal discovery algorithm that can be utilized in this setting?”*
>
> As far as we are aware this does not yet exist, so for now we use a standard conditional independence test. This is an exciting area for new research however!
>
> **Q:** *“The result with direct independence testing is better than those with causal discovery algorithm. Would it be desirable to employ some kind of “optimism in the face of uncertainty” for a “separating set”?”*
>
> **A:** We think exactly the reason direct independence testing performs better is that it is more optimistic with regard to separating sets. In sparse graphs, this seems to be a sensible strategy.

---

> > ### Comment · Reviewer_nePs · 2021-12-12
> > **.**
> >
> > Thanks for the response. It seems that there are several interesting research directions.

---

### Official Review · Reviewer_7S3C · 2021-11-22

**Confidence:** 3
**Overall Score:** 6

**Main Review:**

In this paper, the authors proposed a new bandit algorithm that can more efficiently estimate the causal effect via learning the causal graphs. The simulation results on the Sachs data show that there is improved cumulative regret compared to classical bandit approaches. I took a brief look at the proof and did not find any problem. Overall I think it is an O.K. paper and may be of interest to the CLeaR audience. Nevertheless, after carefully reviewed this manuscript, I still have the following questions:

1. Theorem 2 only shows that there is variance reduction by given a valid separating set. However, in practice all the separating sets are learned from the algorithm and the causal discovery algorithm may not be guaranteed to find the valid set for separation. Is there any theoretical analysis to leverage the estimation errors from causal discovery algorithms into the regret or variance bound?

2. In Section 5.2, the authors showed improved cumulative regret bound on Sachs data set. This means that when the causal graph is easy to learn, the proposed method can achieve a relatively nicer performance compared to orthodox bandit algorithms. I wonder how the performance would be when we instead apply it on a data set with more complicated causal graph. For example, when it is applied to a data with a dense causal graph or with a causal graph containing latent confounders, would it still perform well than the ordinary bandit methods? Or it would be the other way around?

Overall I think it is an O.K. paper, but considering the problems on theoretical analysis and real world applications mentioned above, the contribution seems rather marginal.

**Summary:**

An O.K. paper

---

> ### Author Response · Authors · 2021-12-03
> **Author response to review**
>
> Dear reviewer,
>
> Thank you for your feedback and the time you spent carefully reviewing this work. To answer your questions:
>
> **Q:** *“Theorem 2 only shows that there is variance reduction by given a valid separating set. However, in practice all the separating sets are learned from the algorithm and the causal discovery algorithm may not be guaranteed to find the valid set for separation. Is there any theoretical analysis to leverage the estimation errors from causal discovery algorithms into the regret or variance bound?”*
>
> **A:** This is an excellent question and something we have also been thinking about. A natural way to derive theoretical guarantees for a causal bandit that relies on causal discovery would start from uniform consistency of the causal graph discovery. This can only be guaranteed under strong, unrealistic conditions (Robins et al., Biometrika 90(3) 2003). On the other hand, we believe that when using our proposed direct testing method, uniform consistency of the conditional independence test is not required to derive theoretical guarantees for a causal bandit that relies on direct testing to estimate separating sets. The reason is that even though an arbitrarily large number of samples is needed to distinguish a weak dependence from an independence, for the causal bandit it is not as crucial to distinguish these two as for estimating the true causal graph. Indeed, a weak dependence will only result in a small error in the estimated payoff (but could lead to a globally different causal graph in the worst case). So we do believe that our work may open up a way to derive theoretical guarantees for causal bandits under mild assumptions. However, deriving such results is nontrivial and we consider this beyond the scope of this work.
>
> **Q:** * “In Section 5.2, the authors showed improved cumulative regret bound on Sachs data set. This means that when the causal graph is easy to learn, the proposed method can achieve a relatively nicer performance compared to orthodox bandit algorithms. I wonder how the performance would be when we instead apply it on a data set with more complicated causal graph. For example, when it is applied to a data with a dense causal graph or with a causal graph containing latent confounders, would it still perform well than the ordinary bandit methods? Or it would be the other way around?”*
>
> As can be seen in Mooij et al. (2020 JMLR), different causal discovery algorithms yield very different outputs on the Sachs dataset, and the resulting causal structures are not simple. If a dataset is generated by a dense SCM, separating sets would be very rare or not exist at all. As can be seen in figure 2(c) in the paper under review, if this is the case, the false positive rate of causal discovery leads to a small decrease in performance compared to the ordinary bandit method. The separating set property is still valid in the presence of confounders or cycles, so these do not pose a problem, as long as a causal discovery method is chosen that can handle these structures.

---

### Official Review · Reviewer_KuUF · 2021-11-23

**Confidence:** 3
**Overall Score:** 7

**Main Review:**

This paper looks good enough to be accepted. That said, the clarity should be improved.

Please reword and be more specific when presenting definitions: “If we do not only observe the rewards but also some additional variables, and the rewards and these variables are generated according to a causal model, then the rewards are no longer independent, and this is called a Causal Bandit problem.”

Be more specific about which previous works you are referring to. “In this case, the **previously cited works** on this topic have shown that one can this additional structure to improve our performance.”

How is the context variable different from the regime indicator variable F_t used by Philip Dawid? https://arxiv.org/pdf/2004.12493 (See section 2). Note that while this reference is relatively new the use of such variables can be found in older papers by Dawid and in Pearl’s Causality book.

In the introduction, the authors state, “we develop a framework for Causal Bandits where everything is learned from scratch.” This is an exaggeration.  Later on in section 3.1 they state, “we introduce a novel information sharing estimator that relies on much less specific knowledge about the causal graph”. Please state explicitly the information being assumed about the causal graph.

Typos and spelling mistakes: “then much less restrictive **then** specifically **focussing** on”

The algorithm is presented in the appendix. It should probably be moved to the main paper.


**Summary:**

Develops an algorithm for discrete causal bandits that does not assume prior knowledge about the causal structure.

---

> ### Author Response · Authors · 2021-12-03
> **Response to review**
>
> Dear reviewer,
>
> Thank you for your positive feedback.  The points of clarity raised will greatly benefit the readability of the work!
>
> To answer your questions:
>
> **Q:** *“How is the context variable different from the regime indicator variable F_t used by Philip Dawid? https://arxiv.org/pdf/2004.12493 (See section 2). Note that while this reference is relatively new the use of such variables can be found in older papers by Dawid and in Pearl’s Causality book.”*
>
> **A:** It is not different and is used here just as a convenient modeling choice. The proper citations will be added.
>
> **Q:** *“In the introduction, the authors state, “we develop a framework for Causal Bandits where everything is learned from scratch.” This is an exaggeration. Later on in section 3.1 they state, “we introduce a novel information sharing estimator that relies on much less specific knowledge about the causal graph”. Please state explicitly the information being assumed about the causal graph.”*
>
> **A:** In the theoretical analysis, a given separating set is assumed. In the presented algorithms and experimental results, however, separating sets are found purely from data and thus the statement in the introduction is not an exaggeration. We will clarify the simplifying assumption used in section 3.1.

---

### Decision · Program_Chairs · 2022-01-12

**Decision:**

Accept (Poster)

**Comment:**

This paper proposes a causal bandit algorithm that does not assume prior knowledge about the causal structure and efficiently estimates the causal effect using a separating set between intervention and reward among different arms. The simulation results show that the proposed algorithm outperforms causal bandit approaches that do not leverage causal information obtainable through data and conditional independence tests.

The reviewers agree that the paper proposes an attractive solution to the causal bandit problem. However, the clarity and exposition of the paper can be improved. Therefore, we recommend the authors take the reviewers' comments to heart and encourage them to incorporate their thoughts in preparing the final version of their manuscript.